# *Cryptosporidium myocastoris* n. sp. (Apicomplexa: Cryptosporidiidae), the Species Adapted to the Nutria (*Myocastor coypus*)

**DOI:** 10.3390/microorganisms9040813

**Published:** 2021-04-12

**Authors:** Jana Ježková, Zlata Limpouchová, Jitka Prediger, Nikola Holubová, Bohumil Sak, Roman Konečný, Dana Květoňová, Lenka Hlásková, Michael Rost, John McEvoy, Dušan Rajský, Yaoyu Feng, Martin Kváč

**Affiliations:** 1Faculty of Agriculture, University of South Bohemia in České Budějovice, Studentská 1668, 37005 České Budějovice, Czech Republic; jezkja@seznam.cz (J.J.); jitka.prediger@gmail.com (J.P.); nikoleto@seznam.cz (N.H.); konecnyroman@centrum.cz (R.K.); rost@jcu.cz (M.R.); 2Institute of Parasitology, Biology Centre of the Czech Academy of Sciences, Branišovská 31, 37005 České Budějovice, Czech Republic; zlatalimpouchova@seznam.cz (Z.L.); casio@paru.cas.cz (B.S.); dana@paru.cas.cz (D.K.); hlaskova.lenka@seznam.cz (L.H.); 3Microbiological Sciences Department, North Dakota State University, 1523 Centennial Blvd, Van Es Hall, Fargo, ND 58102, USA; john.mcevoy@ndsu.edu; 4Faculty of Forestry, Technical University in Zvolen, 960 01 Zvolen, Slovakia; dusan.rajsky@gmail.com; 5Key Laboratory of Zoonosis of Ministry of Agriculture, College of Veterinary Medicine, South China Agricultural University, Guangzhou 510642, China; yyfeng@scau.edu.cn; 6Guangdong Laboratory for Lingnan Modern Agriculture, Guangzhou 510642, China

**Keywords:** adaptation, prevalence, biology, course of infection, infectivity, oocyst size, phylogeny, parasite

## Abstract

*Cryptosporidium* spp., common parasites of vertebrates, remain poorly studied in wildlife. This study describes the novel *Cryptosporidium* species adapted to nutrias (*Myocastor coypus*). A total of 150 faecal samples of feral nutria were collected from locations in the Czech Republic and Slovakia and examined for *Cryptosporidium* spp. oocysts and specific DNA at the *SSU*, actin, *HSP*70, and *gp60* loci. Molecular analyses revealed the presence of *C. parvum* (*n* = 1), *C. ubiquitum* subtype family XIId (*n* = 5) and *Cryptosporidium myocastoris* n. sp. XXIIa (*n* = 2), and XXIIb (*n* = 3). Only nutrias positive for *C. myocastoris* shed microscopically detectable oocysts, which measured 4.8–5.2 × 4.7–5.0 µm, and oocysts were infectious for experimentally infected nutrias with a prepatent period of 5–6 days, although not for mice, gerbils, or chickens. The infection was localised in jejunum and ileum without observable macroscopic changes. The microvilli adjacent to attached stages responded by elongating. Clinical signs were not observed in naturally or experimentally infected nutrias. Phylogenetic analyses at *SSU*, actin, and *HSP*70 loci demonstrated that *C. myocastoris* n. sp. is distinct from other valid *Cryptosporidium* species.

## 1. Introduction

*Cryptosporidium* is a protist genus that infects the gastrointestinal and respiratory tract of vertebrate hosts [1]. Cryptosporidiosis, the disease caused by members of this genus, frequently results in diarrhoea, which can be severe and fatal [2]. However, many species and genotypes of *Cryptosporidium*, particularly those infecting wild animals, do not cause clinical signs [3,4]. Genetic and biological studies have shown a high diversity within the genus *Cryptosporidium*, with much of this diversity observed in wildlife hosts [5,6,7,8,9,10]. To date, 47 valid species [11,12,13] and more than 100 genotypes, which are distinguished from valid species on the basis of molecular differences and probably represent separate species, have been described [2]. However, much remains to be discovered about the diversity of the genus *Cryptosporidium* and its host range.

Concerning host specificity, some species of *Cryptosporidium* have a broad host range (e.g., *C. parvum, C. meleagridis*, *C. baileyi* and *C. ubiquitum*), whereas others are restricted to a narrow group of hosts (e.g., *C. muris* and *C. andersoni*) or a single host (e.g., *C. wrairi*) [14].

The nutria (*Myocastor coypus*), also called coypu, is a native rodent of South America and has been introduced to several countries through meat and fur-farming [15]. In many of the regions where farming of nutria is popular, escaped individuals have established a local feral population. Feral nutrias occur on all continents, and they are included as one of the Top 100 Invasive Alien Species of Union Concern in Europe [16]. Parasites of wild, feral, and farmed nutrias are poorly studied [17,18,19], and there have been only four reports of *Cryptosporidium* infections.

*Cryptosporidium* sp., which based on oocyst size were reported as *C. parvum*, were first identified in the faeces of farmed nutrias in Poland [20]. The occurrence of *Cryptosporidium* sp. in wild nutrias was first described in Argentina in 2012 [15], but similarly to Pavlásek and Kozakiewicz [20], isolates were not genetically characterised and therefore the species remains unknown. *Cryptosporidium* copro-antigens were not detected in faeces from feral nutrias from Italy, and a study in the Czech Republic did not find *Cryptosporidium* DNA in faeces from farmed nutrias [18,21]. However, we later identified *C. ubiquitum* and the novel *Cryptosporidium* sp. coypu genotype in feral nutrias in Slovakia.

Building on our earlier findings, we performed a comprehensive study of *Cryptosporidium* in feral nutrias in the Czech Republic and Slovakia. We obtained an isolate of *Cryptosporidium* sp. coypu genotype and determined its biological properties, including oocyst size, host specificity, course and location of infection, and pathogenicity. Based on these data and data from our previous study, we conclude that the *Cryptosporidium* sp. coypu genotype is genetically and biologically distinct from valid *Cryptosporidium* species and is adapted to nutrias. We propose that it be named as a new species, *Cryptosporidium myocastoris* n. sp.

## 2. Materials and Methods

### 2.1. Area and Specimens Studied

Faecal samples from feral nutrias were collected from 7 and 11 localities in the Czech Republic and Slovakia, respectively, during the period 2016–2019 (Figure 1). Faecal samples were individually collected from the rectum of hunted nutria post-mortem or from the ground on riverbanks, placed into a sterile plastic tube, and delivered to the laboratory for processing. A faecal smear was prepared, stained with aniline–carbol–methyl violet (ACMV), and examined for the presence of *Cryptosporidium* spp. oocysts using light microscopy [22]. Infection intensity was expressed as the number of oocysts per gram of faeces (OPG) [23]. The OPG was estimated from the total number of oocysts on the slide and the mass of the faecal smear (approximately 0.015 g). Total genomic DNA (gDNA) was isolated and screened for the presence of *Cryptosporidium*-specific DNA by PCR/sequence analysis of the small subunit ribosomal RNA (*SSU*) gene (described below).

### 2.2. Molecular Study

Total gDNA was extracted from 200 mg of faecal samples and 100 mg of tissue specimens using a GeneAll^®^ Exgene^TM^ Stool DNA mini Kit (GeneAll Biotechnology, co., Ltd.; Seoul, South Korea) and DNeasy Blood & Tissue kits (QIAGEN, Hilden, Germany), respectively, followed by bead disruption for 60 s at 5.5 m/s using 0.5 mm glass beads in a FastPrep^®^-24 Instrument (MP Biomedicals, Santa Ana, CA, USA). The acquired gDNA was stored at −20 °C.

Nested PCR protocols were used to amplify a partial sequence of the *SSU*, actin, 70 kDa heat shock protein (*HSP*70), and 60 kDa glycoprotein (*gp60*) genes using previously published protocols [24,25,26,27,28,29]. For the *SSU* fragment, the primers 5′TTCTAGAGCTAATACATGCG3′ and 5′CCCATTTCCTTCGAAACAGGA3′ were used in the primary reaction, and the primers 5′GGAAGGGTTGTATTTATTAGATAAAG3′ and 5′AAGGAGTAAGGAACAACCTCCA3′ were used in the secondary reaction. For the actin fragment, the primers 5′ATCRGWGAAGAAGWARYWCAAGC3′ and 5′AGAARCAYTTTCTGTGKACAAT3 were used in the primary reaction, and the primers 5′CAAGCWTTRGTTGTTGAYAA3′ and 5′TTTCTGTGKACAATWSWTGG3′ were used in the secondary reaction. For the HSP70 fragment, the primers 5′GCTCGTGGTCCTAAAGATAA3′ and 5′ACGGGTTGAACCACCTACTAAT3′ were used in the primary reaction, and the primers 5′ACAGTTCCTGCCTATTTC3′ and 5′GCTAATGTACCACGGAAATAATC3′ were used in the secondary reaction. For the *gp60* fragment, the primers 5′ATAGTCTCCGCTGTATTC3′ and 5′GGAAGGAACGATGTATCT3′ were used in the primary reaction and the primers 5′TCCGCTGTATTCTCAGCC3′ and 5′GCAGAGGAACCAGCATC3′ were used in the secondary reaction.

Some PCR conditions were modified from the original publications, as previously reported [11]. Molecular grade water and DNA of *C. occultus* were used as negative and positive controls, respectively, for the amplification of *SSU*, actin, and *HSP70* genes. DNA of *C. hominis* subtype family Ib was used as a positive control for amplification of the *gp60* fragment of *C. parvum* and *Cryptosporidium myocastoris* n. sp. DNA of *C. ubiquitum* subtype family XIIa was used as a positive control for the amplification of *C. ubiquitum*. The secondary PCR products were separated by electrophoresis on an agarose gel, stained with ethidium bromide and visualised under UV illumination. All amplicons were purified using the GenElute™ Gel Extraction Kit (Sigma-Aldrich, St. Louis, MO, USA) and directly sequenced using the secondary PCR primers at Eurofins (Prague, Czech Republic).

Chromatogram analysis was performed using Chromas Pro 2.1.4 (Technelysium, Pty, Ltd., South Brisbane, Australia), and sequences were verified by BLAST analysis (https://blast.ncbi.nlm.nih.gov/Blast.cgi; accessed on 20 February 2021). The sequences obtained in this study and reference sequences obtained from GenBank were aligned using the MAFFT version 7 online server (http://mafft.cbrc.jp/alignment/server/; accessed on 20 February 2021) using the E-INS-i multiple alignment method. The alignments were manually trimmed and edited in BioEdit v.7.0.5 [30]. Phylogenetic analysis was performed using the maximum likelihood (ML) method, using evolutionary models selected by MEGAX [31]. Bootstrap supports were calculated from 1000 replications. Phylogenetic trees were produced by MEGAX and further edited for visualisation purposes with Corel DrawX7 (Corel Corporation, Ottawa, Ontario, Canada). Species-specific divergences were identified from proportional distances (%) which were calculated in the program Geneious v11.0.3 [32] based on the *SSU*, actin, and *HSP70* datasets of all sequences under study. All nucleotide sequence data were deposited in GenBank, Accession Numbers (Acc. nos.): MW274645-MW274661 for *SSU*, MW280959-MW280975 for actin, MW280976-MW280992 for *HSP*70, and MW280993-MW281009 for *gp60*.

### 2.3. Source of Oocysts of Cryptosporidium myocastoris n. sp.

Oocysts of *C. myocastoris* n. sp., recovered from naturally infected nutrias (isolate 31132) using sucrose [33] and caesium chloride gradient centrifugation [34], were used to experimentally infect a *Cryptosporidium*-negative nutria (nutria 0) and for morphometric study (described below).

### 2.4. Animals for Transmission Studies

Six adult nutrias (nutria 0 and nutrias 1–5; *Myocastor coypus*), five one-week- and eight-week-old gerbils (*Meriones unguiculatus*), five one-week- and eight-week-old SCID mice (*Mus musculus*; strain C.B-17), five one-week- and eight-week-old BALB/c mice, and five one-day-old chickens (*Gallus gallus* f. *domestica*) were used for transmission studies. Three animals from each group were used as negative controls. All nutrias used for the transmission study were screened daily for the presence of specific DNA and oocysts of *Cryptosporidium* for two weeks prior to experimental inoculation. Mice and gerbils, which were bred under laboratory conditions, were screened for the presence of specific DNA and oocysts of *Cryptosporidium* one week prior to experimental inoculation. One-day-old chickens, which were hatched under laboratory conditions, were screened on the day of hatching.

### 2.5. Animal Care

To prevent environmental contamination with oocysts, mice and gerbils were individually housed in ventilated cages (Tecniplast, Buguggiate, Italy), and chickens were housed in plastic boxes that were disinfected at 80 °C for one hour before being used. Nutrias were kept in boxes with secured walls. The individual boxes were disinfected with pressurized steam before use. Cages and boxes were sized in accordance with European and Czech Republic regulations on the protection of animals against cruelty. An external source of heat was used in the first five days for chickens. Each animal was supplied with a sterilized diet and sterilized water ad libitum. Animal keepers wore disposable protective equipment during care of the animals. Woodchip and straw bedding and disposed protective clothing were removed from the experimental room and incinerated. All experimental procedures complied with the law of Czech Republic (Act No. 246/1992 Coll., on the protection of animals against cruelty). The study design was approved by ethical committees at the Biology Centre of CAS, the State Veterinary Administration, and the Central Commission for Animal Welfare under Protocol No. 35/2018 and 60/2019.

### 2.6. Design of Transmission Studies

Nutria 0 was inoculated orally with 5,000 purified oocysts of *C. myocastoris* n. sp. recovered from naturally infected nutria (described above) and suspended in 500 μL of distilled water. Oocysts of *C. myocastoris* n. sp., recovered from experimentally infected nutria 0, were molecularly characterised and used to infect other experimental animals (above). Other animals were inoculated with 20,000 purified oocysts of *C. myocastoris* n. sp. suspended in 500 μL (nutria), 200 μL (mice, gerbils, and chickens) of distilled water. Animals serving as negative controls were inoculated orally with 500 μL (nutria), 200 μL (mice and gerbils) or 20 μL (chickens) of distilled water. Faecal samples from each animal were screened daily for the presence of *Cryptosporidium* oocysts (ACMV staining) and *Cryptosporidium*-specific DNA (*SSU* gene amplification and sequencing). All experiments were terminated 30 days post-infection (DPI). The course of infection was evaluated based on the presence of *C. myocastoris* n. sp. specific DNA and the number of oocyst per gram of faeces as previously described by Kváč et al. [23]. Consistency and colour of faeces and health status were determined daily for each sample and animal, respectively. From each experimentally infected animal, PCR-positive samples from the beginning, middle, and end of the infection were additionally examined at the actin, *HSP70* and *gp60* genes to verify the identity of the *C. myocastoris* n. sp. with the inoculum and the original isolate.

### 2.7. Morphological Evaluation

The oocysts of *Cryptosporidium myocastoris* n. sp., which originated from naturally infected nutrias (isolates 31132 and 31459, Table 1) and experimentally infected nutrias (nutria N0 and nutria N1), were purified using sucrose and caesium chloride gradient centrifugation and examined using differential interference contrast (DIC) microscopy, bright field microscopy following ACMV and modified Ziehl-Neelsen (ZN) staining [35], and fluorescence microscopy following labelling of the *Cryptosporidium* oocyst wall with genus-specific FITC-conjugated antibodies (IFA; *Cryptosporidium* IF Test, Cryptocel, Cellabs Pty Ltd., Brookvale, Australia). Images of oocysts were collected using an Olympus Digital Colour camera (DP73) and Olympus cell SensEntry 2.1 software (Olympus Corporation, Shinjuku, Tokyo, Japan). Length and width of oocysts (*n* = 30) from naturally and experimentally infected nutrias were measured under DIC at 1000× magnification. The length-to-width ratio was calculated for each oocyst. As a control, the morphometry of *C. parvum* from naturally infected calfs (*Bos taurus*; *n* = 30) and *C. ratti* from experimentally infected rats (*Rattus novegicus*; *n* = 30) was used. Photomicrographs of *Cryptosporidium myocastoris* n. sp. oocysts under DIC, ACMV, ZN and IFA are part of this publication, and have been deposited as a phototype at the Institute of Parasitology, Biology Centre of the Czech Academy of Sciences, Czech Republic.

### 2.8. Clinical and Pathomorphological Examinations

A complete necropsy of two animals from each experimental group was performed at 10 and 20 DPI. Tissue specimens (oesophagus, stomach, duodenum, proximal, central and distal jejunum, ileum, caecum, colon, liver, spleen, kidney, bladder, and lung) from each animal were obtained using different sterile dissection tools for each location. Specimens from each organ were collected for PCR/sequencing, histology, and scanning electron microscopy (SEM). Histology sections were processed as reported by Kváč and Vítovec [36], and SEM sections were processed as described by Holubová et al. [11]. Histology sections were stained with haematoxylin and eosin (HE) and periodic acid–Schiff (PAS), examined at 100–400× magnification and documented using Olympus cell Sens Entry 2.1 (Olympus Corporation, Shinjuku, Tokyo, Japan) equipped with a digital camera (Olympus DP73). Specimens for SEM were examined using a JEOL JSM-7401F-FE scanning electron microscope equipped with a digital camera ETD Detector A PRED (Termo Fisher Scientific, Waltham, MA, USA).

### 2.9. Statistical Analysis

Differences in oocyst sizes were tested using Hotelling’s multivariate version of the 2 sample *t*-test, *package ICSNP: Tools for Multivariate Nonparametrics* [37] in R 4.0.0. [38]. The hypothesis tested was that two-dimensional mean vectors of measurement are the same in the two populations being compared.

## 3. Results

In total, 72 and 78 faecal samples were examined from the Czech Republic and Slovakia, respectively (Table 1). None of the faecal samples had a consistency that would indicate diarrhoea. Examination of faecal smears revealed the presence of *Cryptosporidium* sp. oocysts in three samples, and infection intensity ranged from 10,000 to 25,000 OPG (Table 1). *Cryptosporidium*-specific DNA was detected in 11 samples by nested PCR targeting the *SSU* gene (Figure 2). From these positive samples, partial sequences of the genes encoding actin, *HSP*70 and *gp60* were amplified/sequenced. ML trees constructed from *SSU,* actin and *HSP70* sequences showed the presence of *C. parvum* (*n* = 1), *C. ubiquitum* (*n* = 5) and *Cryptosporidium myocastoris* n. sp. (*n* = 5). *Cryptosporidium myocastoris* n. sp., previously known as the *Cryptosporidium* coypu genotype, is described in detail as a new species later in this publication (Figure 2, Figure 3 and Figure 4). There was no intraspecies variability in *SSU*, actin, and *HSP70* sequences from this study. Subsequent subtyping of *C. parvum* and *C. ubiquitum* at the *gp60* gene showed the presence of subtype families IIaA16G1R1 and XIId, respectively (Table 1, Figure 5). All *gp60* sequences from *C. ubiquitum* were identical (Figure 5). Two novel subtype families, which we have named XXIIa (*n* = 2) and XXIIb (*n* = 3), were detected within *C. myocastoris* n. sp. (Table 2, Figure 5).

Oocysts of *C. myocastoris* n. sp. recovered from naturally infected nutria (isolate 31132) were infectious for *Cryptosporidium*-free farmed nutria (nutria N0), which shed oocysts that were genetically and morphometrically identical to the inoculum, from five DPI (Table 2; Figure 2, Figure 3, Figure 4 and Figure 5). 

Oocysts of *C. myocastoris* n. sp. from nutria N0 were infectious for five farmed nutrias (nutrias N1–N5), but not for one-week- and eight-week-old BALB/c and SCID mice, gerbils, or one-day-old chickens. All groups of nutrias, mice, gerbils, and chickens used as negative controls remained uninfected. Experimentally infected nutrias N1–N5 started to shed oocysts of *C. myocastoris* n. sp., detectable by light microscopy and PCR, at 5–6 DPI, and all animals remained infected until the end of the experiment (Figure 6). The infection intensity ranged from 2000 to 62,000 OPG. The highest infection intensity, 20,000–62,000 OPG, was observed from 6 to 11 DPI. Beginning on day 12 post-infection, all animals shed fewer than 10,000–15,000 OPG (Figure 6). None of the animals showed signs of cryptosporidiosis, and faecal consistency was appropriate to the age of the animal and the food intake. *Cryptosporidium myocastoris* n. sp. DNA and *Cryptosporidium* developmental stages were detected exclusively in posterior jejunum and ileum (Figure 7 and Figure 8). Histology and SEM showed low infection intensity, with one or two developmental stages typically observed on an isolated villus in the posterior jejunum and ileum (Figure 7 and Figure 8). This low infection intensity was consistent throughout the posterior jejunum and ileum. The brush border microvilli adjacent to attached developmental stages responded by elongation (Figure 8). The area of elongated microvilli increased with the size of the developmental stage, and the microvilli were elongated by up to 2 μm (Figure 8). The lamina propria in the jejunum and ileum was occasionally observed to be slightly oedematous, but these changes were probably not related to the *Cryptosporidium* infection.

Oocysts of *C. myocastoris* n. sp. recovered from experimentally infected nutrias (nutria N0 and N1) were morphometrically identical to those recovered from naturally infected nutrias (isolates 31132 and 31459; Table 2).


***Taxonomic summary:***

**Family Cryptosporidiidae Léger, 1911**

**Genus *Cryptosporidium* Tyzzer, 1907**
***Cryptosporidium myocastoris*** n. sp.**Synonym:***Cryptosporidium* sp. coypu genotype ex *Myocastor coypus* of Kváč et al. [39], Slovakia.**Type-host:***Myocastor coypus* (Molina, 1782), nutria.**Other natural hosts:** No other natural hosts are known.**Type-locality:** Dunajská Streda (N 47°55.90470′, E 17°28.42662′), Slovakia.**Other localities:** Lanžhot (N 48°43.41558′, E 16°58.30782′), Czech Republic; Šaľa (N 48°9.08273′, E 17°52.49152′), Slovakia.**Type-material:** Faecal smear slides with oocysts stained by ACMV and ZN staining (nos. MV1/31132 and ZN2/31132); scanning electron microscopy specimens of infected jejunum (no. SEM23/2017) and ileum (no. SEM27/2017); histological sections of infected jejunum (no. H23/2017) and ileum (no. H27/2017); gDNA isolated from faecal samples of naturally (isolate 31132) and experimentally (isolate 32235) infected nutrias; gDNA isolated from jejunum and ileum of experimentally infected nutrias (isolates 32235 and 32236). All specimens are deposited at the Institute of Parasitology, Biology Centre of the Czech Academy of Sciences, Czech Republic.**Site of infection:** Posterior jejunum and ileum (present study, Figure 7 and Figure 8).**Distribution:** As *Cryptosporidium* sp. coypu genotype ex *Myocastor coypus*: Slovakia [39].**Prepatent period:***Myocastor coypus* 5–6 days (present study).**Patent period:** At least 30 days in experimentally infected nutrias (*Myocastor coypus*; present study).**Representative DNA sequences:** Representative nucleotide sequences of *SSU* [MW274649], actin [MW280963], *HSP*70 [MW280980] and *gp60* [MW280997 and MW280994] genes were saved in the GenBank database.**ZooBank registration:** To comply with the regulations set out in Article 8.5 of the amended 2012 version of the International Code of Zoological Nomenclature (ICZN) [40], details of the new species have been submitted to ZooBank. The Life Science Identifier (LSID) of the article is urn:lsid:zoobank.org:pub:FCAD0ED3-2DD0-4A79-93DD-D0C206EC6ACF. The LSID for the new name *Cryptosporidium myocastoris* n. sp. is urn:lsid:zoobank.org:act:E447F777-5495-4613-8447-D015339F6B32.**Etymology:** The species name *myocastoris* is derived from the Latin noun myocastor, meaning nutria.**Description:** Oocysts of *C. myocastoris* n. sp. (isolate 31132) are spherical, measuring 4.8–5.2 × 4.7–5.0 (5.02 ± 0.13 × 4.85 ± 0.10) with a length-to-width ratio of 1.00–1.08 (1.04 ± 0.02) (Figure 9). The oocyst wall is smooth and colourless (Figure 9a). The oocyst residuum is composed of numerous small granules and one spherical globule is clearly visible; a suture is not noticeable. Four sporozoites are clearly visible within oocysts. The morphology and morphometry of other developmental stages is unknown.

**Remarks.** Oocysts of *C. myocastoris* n. sp. are well stained by ACMV and ZN staining methods, similarly to other *Cryptosporidium* spp. (Figure 9b,c), and their oocyst walls cross-react with immunofluorescence reagents developed primarily for *C. parvum* (Figure 9d). Oocysts from naturally and experimentally infected nutrias did not differ in size (T^2^ = 0.16, *df*_1_ = 2, *df*_2_ = 121, *p* = 0.8506; Table 2). Oocysts of *C. myocastoris* n. sp. are smaller than those of *C. parvum* (T^2^ = 33.11, *df*_1_ = 2, *df*_2_ = 48.15, *p* = *p* < 0.001) and *C. ratti* (T^2^ = 33.22, *df*_1_ = 2, *df*_2_ = 45.96, *p* < 0.001), but these differences are not of practical significance for identification (Table A1 in Appendix A). *Cryptosporidium myocastoris* n. sp. can be differentiated genetically from other *Cryptosporidium* species based on sequences of *SSU*, actin, and *HSP*70 genes. At the *gp60* locus, *C. myocastoris* n. sp. develops two well-supported clades. Pairwise distances between *Cryptosporidium myocastoris* n. sp. and the selected closest and furthest *Cryptosporidium* species at *SSU*, actin, and *HSP*70 genes are shown in Table A2.

## 4. Discussion

The work presented here represents the most comprehensive study to date on *Cryptosporidium* infecting feral nutrias, and a novel, nutria-adapted *Cryptosporidium* species is described. The prevalence of *Cryptosporidium* in nutrias was relatively low (7.3%). In a study of 108 wild nutrias in Argentina, the prevalence of *Cryptosporidium* infection was similarly low (3.7%), although oocysts were detected by microscopy, which is less sensitive than PCR [15]. The prevalence of *Cryptosporidium* in other aquatic rodents also appears to be low. A study of 145 capybaras (*Hydrochoerus hydrochaeris*) in Brazil reported a prevalence of 5.5% [41]. Zhou et al. [42] did not detect *Cryptosporidium* in any of the 84 North American beavers (*Castor canadensis*) they tested in the United States, but they did report that almost 12% of 237 muskrats (*Ondatra zibethicus*) were positive. Paziewska et al. [43] reported that 7.7% of 22 European beavers (*Castor fiber*), sampled in Poland, had *Cryptosporidium* antigen in their faeces. Considering that *Cryptosporidium* infections are frequently associated with transmission through contaminated water [44,45], the low prevalence in nutrias and other water rodents might be surprising. Future studies should address how *Cryptosporidium* spp. are transmitted in aquatic mammals: is waterborne transmission important, or is it some other route such as contaminated food or direct contact among individuals? In this context, it is noteworthy that *C. myocastoris* n. sp. has not been reported in any of the studies on the occurrence of *Cryptosporidium* spp. in surface water [46]. This absence may be explained by (i) the low prevalence within the population of nutrias together with the low infection intensity; or (ii) the limited number of studies reporting water contamination in the areas where nutrias occur.

This is the first study to genotype *Cryptosporidium* from nutrias. Martino et al. [15] found *Cryptosporidium* sp. oocysts (4 to 4.5 × 4.0 μm) in faecal samples from the colon and rectum of wild nutrias in Argentina. An earlier study reported *C. parvum* in nutrias based on oocyst morphology [15,20]. Given that *C. parvum* has a broad host specificity, is reported infrequently in wildlife species [5,6,47,48,49,50], and was found in a single nutria in the present study, it is possible that Pavlásek and Kozakiewicz [20] correctly identified the species. In support, they found that oocysts from naturally infected nutria were infectious for four-day-old laboratory mice under experimental conditions, a characteristic of *C. parvum* but not *C. myocastoris* (as we have shown in the present study). However, based on our current knowledge, oocyst morphology cannot reliably distinguish among intestinal species of *Cryptosporidium* [51], and the size of oocysts reported by Pavlásek and Kozakiewicz [20] (5.0 × 4.75 µm) is similar to *C. myocastoris* n. sp. (5.02 × 4.85 μm) and *C. ubiquitum* (5.04 × 4.66 μm), two species that we found to be more prevalent than *C. parvum* (5.19 × 4.9 μm) in nutrias. 

The finding of *C. ubiquitum* and *C. myocastoris* n. sp. in several animals at different localities suggests that these species are common in feral nutrias. *Cryptosporidium ubiquitum*, a species with broad host specificity, has been reported in domestic and wild ruminants, rodents, carnivores, and human and non-human primates [52,53,54,55,56]. The *C. ubiquitum gp60* subtype family XIId in nutrias has been reported in humans, macaques, red deer, raccoons, woodchucks, chinchillas, and mink [18,29,56,57,58,59], suggesting that it is broadly specific. In contrast, *C. myocastoris* n. sp. appears to have a narrow host specificity. The origin of *C. myocastoris* n. sp. in nutria in central Europe is difficult to elucidate without further studies, but it may have been introduced into Europe with imported nutrias, similarly to the *Cryptosporidium* skunk genotype, which was likely introduced to Europe with eastern grey squirrels [60]. The specificity of *C. myocastoris* n. sp. for nutrias is supported by its presence in geographically isolated nutrias, its infectivity for nutrias under experimental conditions, and the absence of any record of this species in any of the thousands of molecular epidemiological studies published in last two decades [46,51,61,62]. For these reasons, it is most likely that nutrias are the major host of *C. myocastoris* n. sp., although we cannot exclude the possibility that other host species have a role as major or minor hosts.

Adaptation to one host species is not unique within the genus *Cryptosporidium*, and several mammalian *Cryptosporidium* spp. have been reported almost exclusively in a single host. Examples include *C. wrairi* and *C. homai* in guinea pigs (*Cavia porcellus*), *C. scrofarum* and *C. suis* in pigs (*Sus scrofa*), *C. bovis* in cattle (*Bos taurus*), *C. occultus* in rats (*Rattus* spp.), and *C. macropodum* in kangaroos [63,64,65,66,67]. The findings of these species in other hosts represent rare cases or mechanical passage rather than host adaptation [68,69,70]. 

The prepatent period (5–6 DPI) is consistent with other intestinal *Cryptosporidium* spp. that are specific for rodents, such as *C. ratti* (4–5 DPI) in rats, *C. alticolis* in voles (3–4 DPI), *C. tyzzeri* in mice (4–7 DPI), or other mammals, such *C. parvum* in calves (2–7 DPI) and *C. scrofarum* in pigs (4–6 DPI) [6,12,63,66,71,72].

Nutrias positive for *C. parvum* or *C. ubiquitum* did not have detectable oocysts in their faeces, suggesting a low level of infection, which is consistent with our previous finding [39]. Three out of five nutrias naturally infected with *C. myocastoris* n. sp. shed oocysts at levels between 10,000 and 20,000 OPG, and similar numbers were detected in experimentally infected nutrias. No macroscopic changes in the intestinal mucosa were observed at necropsy of *C. myocastoris* positive nutrias. The low level of oocyst shedding was consistent with the intensity of developmental stages detected in the intestinal epithelium and the absence of clinical symptoms. A similar relationship between intestinal involvement and oocyst shedding has been observed in other *Cryptosporidium* spp. infecting the small intestine [6,12,73,74,75,76]. Examination by scanning electron microscopy shows the elongation of the microvilli around the *C. myocastoris* developmental stages, which has also been previously observed in SCID mice infected with *C. parvum* [77]. Borowski et al. [78] reported the elongation of microvilli on the gliding trails of *C. parvum* sporozoites between an excysted oocyst and newly formed trophozoites. While the extending of microvilli in Borrowski’s study was up to 15 µm, we observed much less extension. Clinical symptoms are rarely, if ever, observed in wild animals infected with host-specific *Cryptosporidium* species [79,80,81].

Equivalently to the host specificity, most *Cryptosporidium* species are characterized by adaptation to a different part of the digestive tract. Gastric *Cryptosporidium* of mammals exclusively infect the glandular part of the glandular stomach, whereas intestinal species are adapted to different parts of the small or large intestine. Similar organ adaptation has been reported in *Eimeria* spp. [82]. AS with *C. ratti* or *C. scrofarum*, the life cycle of *C. myocastoris* n. sp. is located in the posterior jejunum and ileum [12,66]. No developmental stages were found in the large intestine where the life cycle of, e.g., *C. occultus*, *C. suis* or *C. ornithophilus* is located [63,65,81].

For species-level differentiation of *Cryptosporidium*, the *SSU* marker has served well for more than 20 years [83]. However, *Cryptosporidium*, similarly to the related apicomplexans *Plasmodium* and *Eimeria,* has divergent paralogous copies of the SSU gene [84,85,86,87,88]. Our previous work has shown that only using sequences of SSU to infer evolutionary relationships of *Cryptosporidium* may lead to erroneous conclusions [5,29,89]. Therefore, it is necessary to use other polymorphic loci, such as *HSP70*, actin, or *COWP* genes, in phylogenetic analyses [5,85]. Although bootstrap support for the *SSU* tree was lower than for the actin and *HSP70* trees in this study, *C. myocastoris* n. sp. formed a separate clade in *SSU*, actin and *HSP70* ML trees, with the most closely related group comprising species such as *C. parvum*, *C. cuniculi*, and *C. wrairi*. At the *SSU* locus, the pairwise distance between *C. myocastoris* n. sp. and *C. parvum* (0.014) or *C. cuniculi* (0.018) is similar to that between *C. apodemi* and *C. occultus* (0.016) or *C. andersoni* and *C. serpentis* (0.019). At the actin locus, the distance between *C. myocastoris* n. sp. and *C. parvum*/*C. cuniculi* (0.039) is similar to that between *C. meleagridis* and *C. erinacei* (0.034). Comparably, at the *HSP70* locus, the distance between *C. myocastoris* n. sp. and *C. parvum*/*C. cuniculi* (0.035/0.031) is similar to the distance between *C. meleagridis* and *C. erinacei* (0.041) or *C. ornithophilus* and *C. avium* (0.035). These results support the genetic uniqueness of *C. myocastoris* n. sp. and their status as a separate species of genus *Cryptosporidium.*

## 5. Conclusions

In summary, the findings that *Cryptosporidium* sp. coypu genotype is genetically distinct from all described *Cryptosporidium* species and specific for nutrias under natural and experimental conditions support its description as a new species, and we propose that it be named *Cryptosporidium myocastoris* n. sp.

## Figures and Tables

**Figure 1 microorganisms-09-00813-f001:**
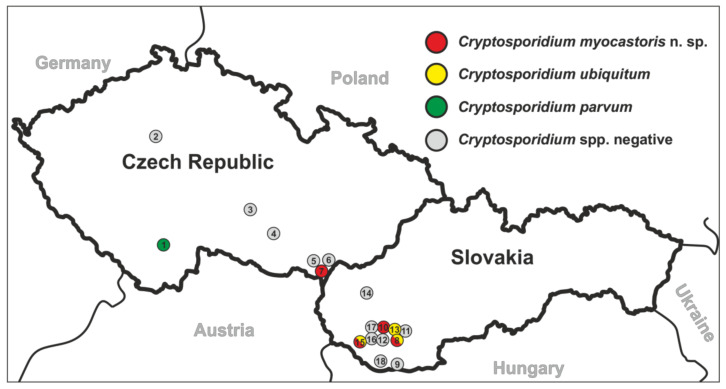
Sampling localities in the Czech Republic and Slovakia. For each site, the number indicates the name of locations (**1**) Planá nad Lužnicí (N 49°21.11527′, E 14°42.08325′); (**2**) Praha (N 50°4.78070′, E 14°24.81330′); (**3**) Jihlava (N 49°23.09785′, E 15°36.37422′); (**4**) Třebíč (N 49°12.58360′, E 15°52.28832′); (**5**) Břeclav (N 48°46.34147′, E 16°52.68835′); (**6**) Týnec (N 48°46.31570′, E 17°0.66778′); (**7**) Lanžhot (N 48°43.41558′, E 16°58.30782′); (**8**) Nové Zámky (N 48°0.76540′, E 18°11.75573′); (**9**) Komárno (N 47°45.04053′, E 18°8.98083′); (**10**) Šaľa (N 48°9.08273′, E 17°52.49152′); (**11**) Doný Ohaj (N 48°4.33237′, E 18°14.81102′); (**12**) Topolníky (N 47°57.61112′, E 17°45.37918′); (**13**) Palárikovo (N 48°2.15190′, E 18°2.70193′); (**14**) Nitrianský Hrádok (N 48°3.62700′, E 18°12.56517′); (**15**) Dunajská Streda (N 47°55.90470′, E 17°28.42662′); (**16**) Vlčny (N 48°2.69967′, E 17°57.72202′); (**17**) Diakovce (N 48°8.02725′, E 17°50.48138′); and (**18**) Lipové (N 47°50.41113′, E 17°51.33057′). The colour indicates the presence of *Cryptosporidium* spp.

**Figure 2 microorganisms-09-00813-f002:**
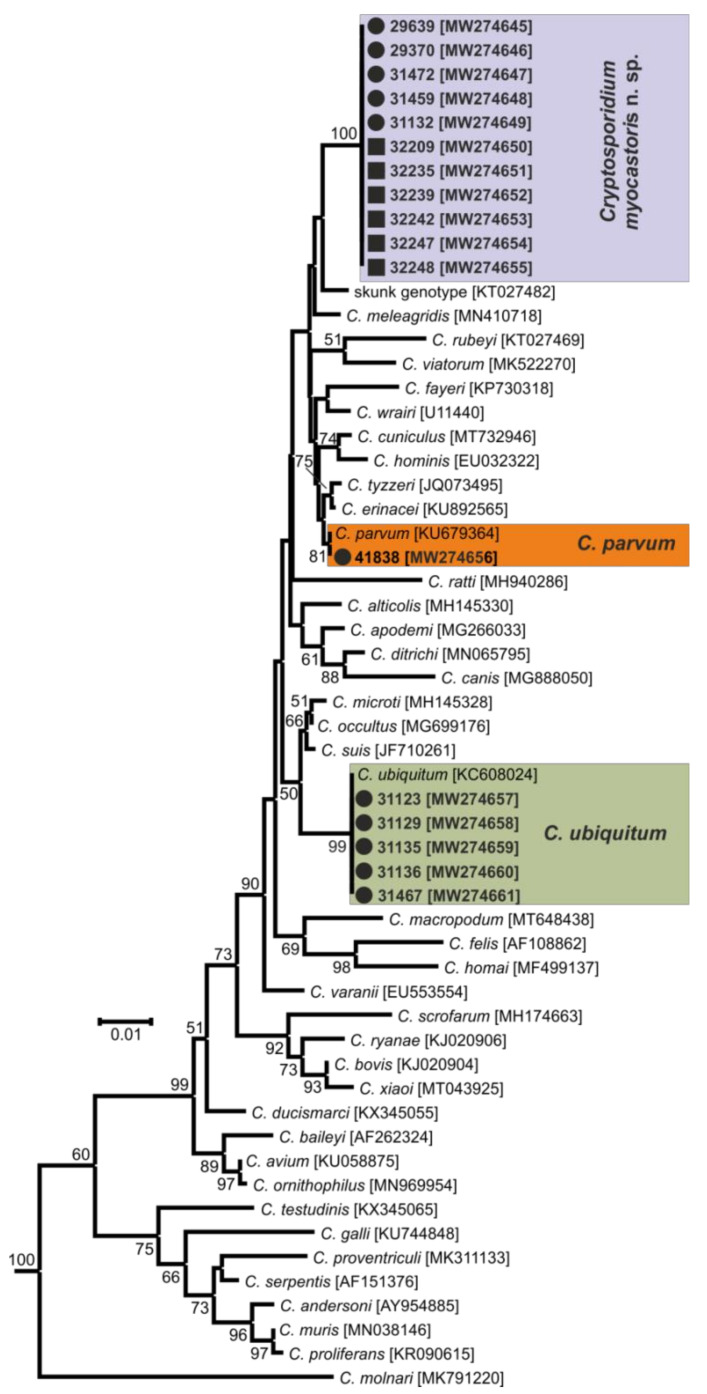
Maximum likelihood tree based on partial small subunit ribosomal RNA gene sequences of *Cryptosporidium* spp., including sequences obtained in this study (bolded and highlighted). The alignment contained 770 base positions in the final dataset. Numbers at the nodes represent the boot strap values with more than 50% boot strap support from 1000 pseudo replicates. The branch length scale bar, indicating the number of substitutions per site, is given in the tree. Sequences from this study are identified by an isolate number (e.g., 32247). Black circles and squares indicate natural and experimental infections, respectively.

**Figure 3 microorganisms-09-00813-f003:**
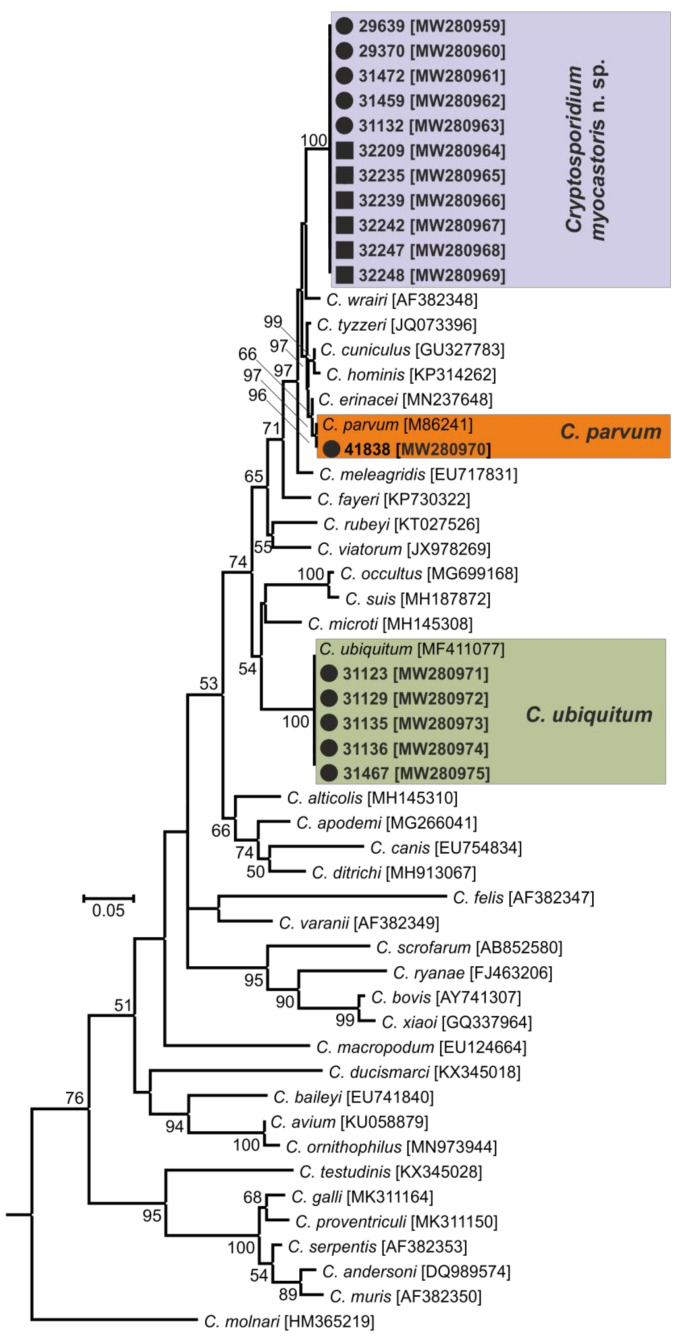
Maximum likelihood tree based on actin gene sequences of *Cryptosporidium* spp., including sequences obtained in this study (bolded and highlighted). The alignment contained 990 base positions in the final dataset. Numbers at the nodes represent the boot strap values with more than 50% boot strap support from 1000 pseudo replicates. The branch length scale bar, indicating the number of substitutions per site, is given in the tree. Sequences from this study are identified by an isolate number (e.g., 32247). Black circles and squares indicate natural and experimental infections, respectively.

**Figure 4 microorganisms-09-00813-f004:**
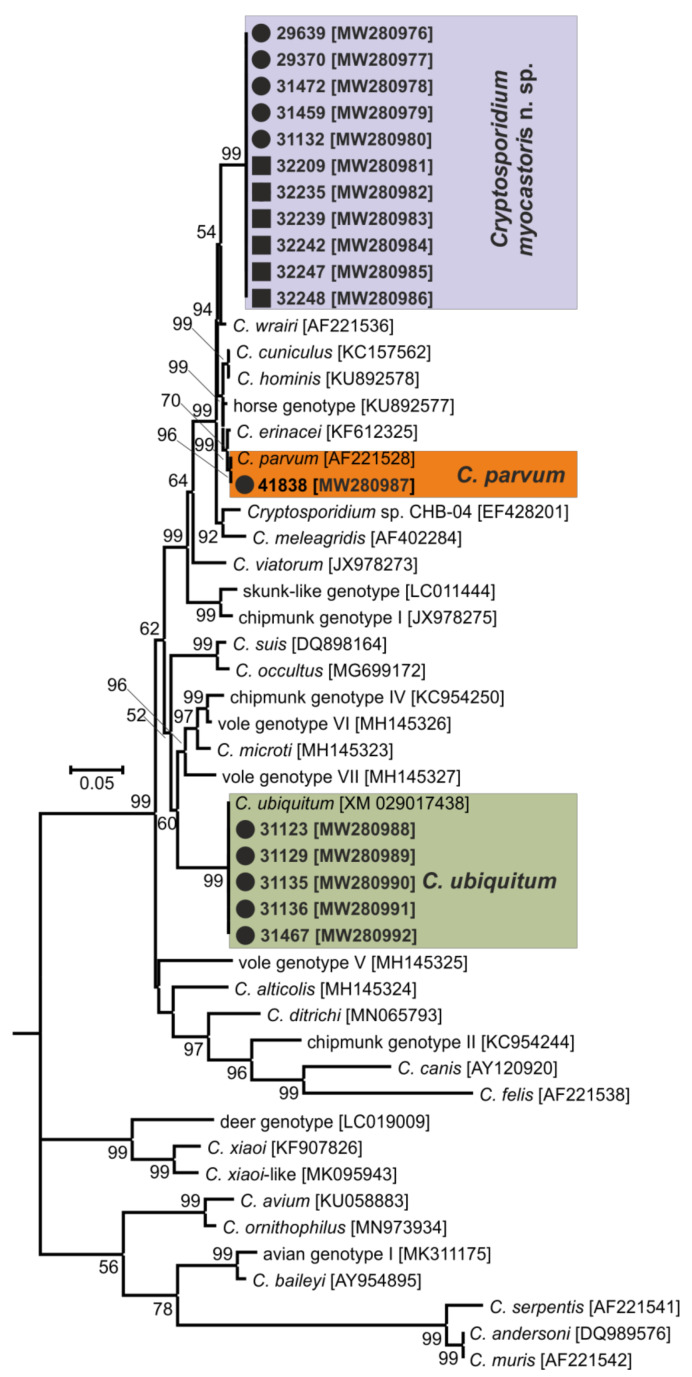
Maximum likelihood tree based on 70 kDa heat shock protein (*HSP70*) gene sequences of *Cryptosporidium* spp., including sequences obtained in this study (bolded and highlighted). The alignment contained 1172 base positions in the final dataset. Numbers at the nodes represent the boot strap values with more than 50% boot strap support from 1000 pseudo replicates. The branch length scale bar, indicating the number of substitutions per site, is given in the tree. Sequences from this study are identified by an isolate number (e.g., 32247). Black circles and squares indicate natural and experimental infections, respectively.

**Figure 5 microorganisms-09-00813-f005:**
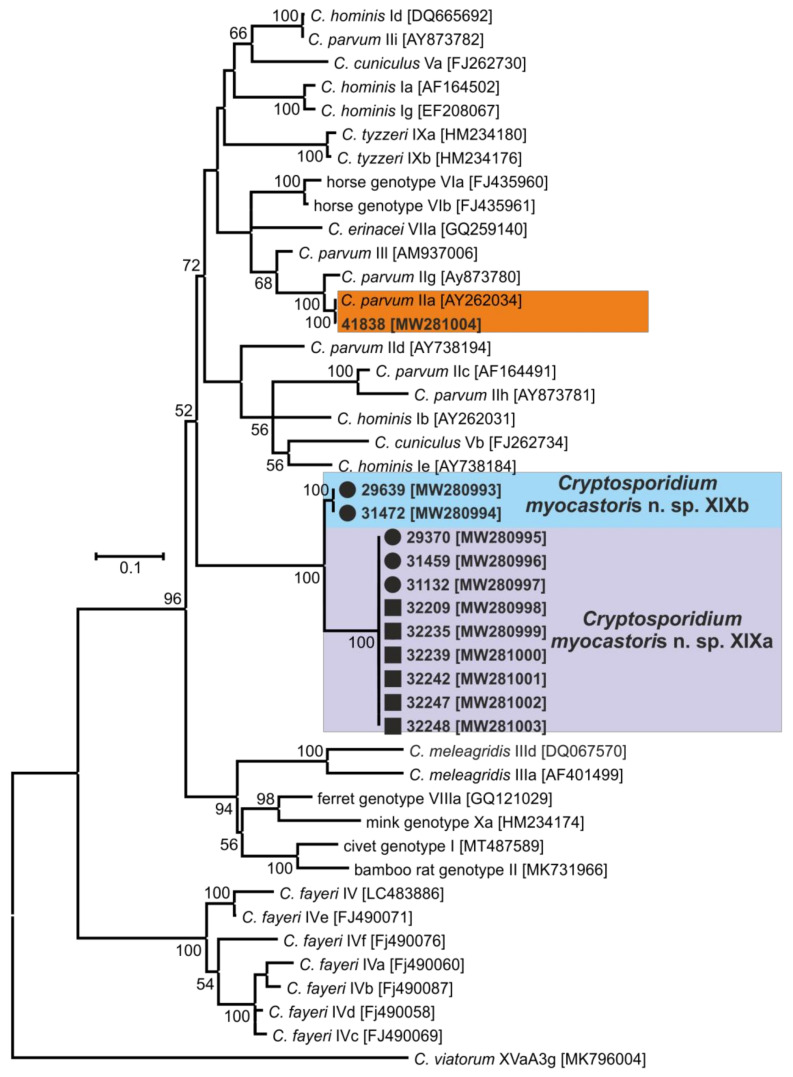
Maximum likelihood tree based on 60 kDa glycoprotein (*gp60*) gene sequences of *Cryptosporidium* spp., including sequences obtained in this study (bolded and highlighted). The alignment contained 1206 base positions in the final dataset. Numbers at the nodes represent the boot strap values with more than 50% boot strap support from 1000 pseudo replicates. The branch length scale bar, indicating the number of substitutions per site, is given in the tree. Sequences from this study are identified by an isolate number (e.g., 32247). Black circles and squares indicate natural and experimental infections, respectively.

**Figure 6 microorganisms-09-00813-f006:**
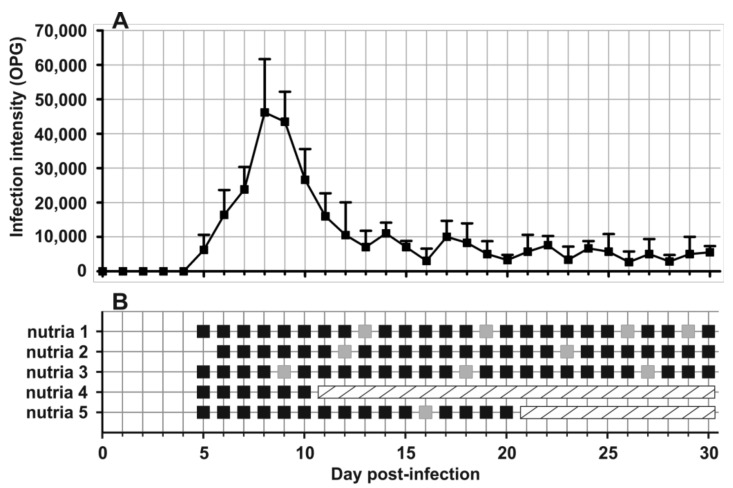
Course of infection of *Cryptosporidium myocastoris* n. sp. in experimentally inoculated nutria (*Myocastor coypu*). (**A**) Infection intensity expressed as number of oocysts per gram of faeces (OPG), and (**B**) detection of oocysts is based on molecular and microscopic examinations of faecal samples. Black squares indicate the presence of oocysts and specific *Cryptosporidium myocastoris* n. sp.; grey squares indicate the detection of specific DNA only without oocyst detection. Hatched rectangles indicate a missing animal due to sacrifice and dissection.

**Figure 7 microorganisms-09-00813-f007:**
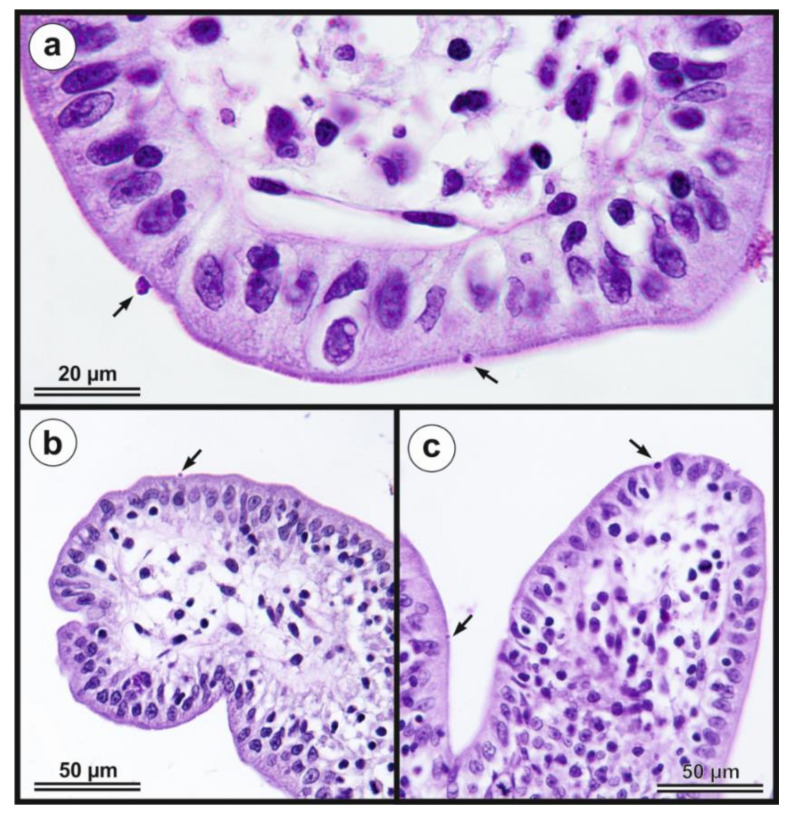
Histological sections stained by periodic acid–Schiff showing developmental stages of *Cryptosporidium myocastoris* n. sp. (arrow) on (**a**) jejunal and (**b,c**) ileal mucosal epithelium in experimentally infected adult nutria (*Myocastor coypu*) which was sacrificed 10 days post infection. Scale bar is included in each figure.

**Figure 8 microorganisms-09-00813-f008:**
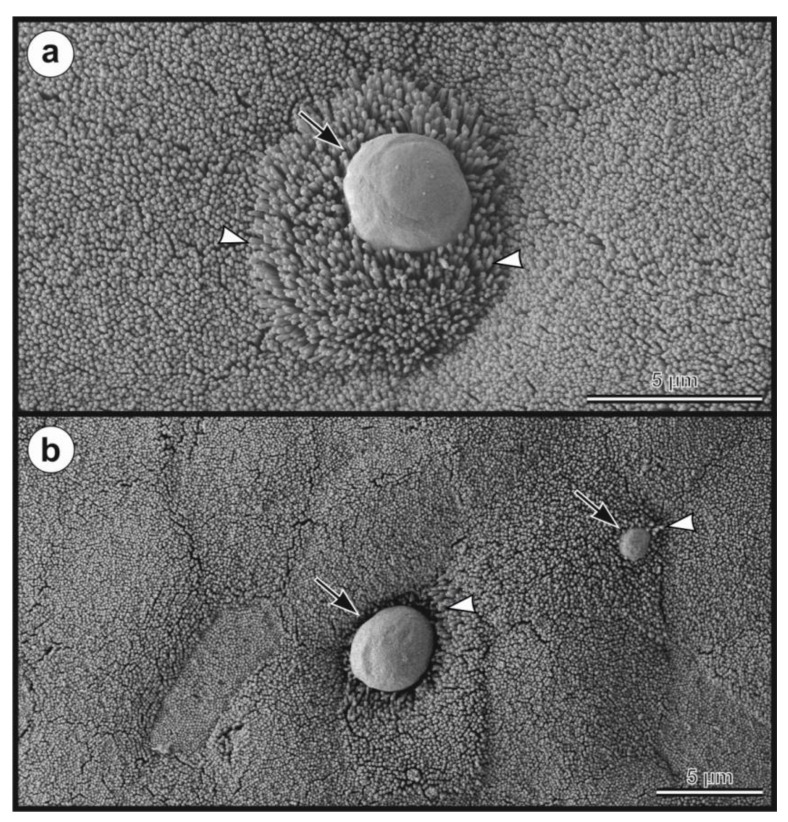
Scanning electron microphotograph showing developmental stages of *Cryptosporidium myocastoris* n. sp. (arrow) on (**a**) jejunal and (**b**) ileal mucosal epithelium in experimentally infected adult nutria (*Myocastor coypu*) which was sacrificed 10 days post infection. Elongation of the microvilli around attached developmental stage (arrowhead). Scale bar included in each figure.

**Figure 9 microorganisms-09-00813-f009:**
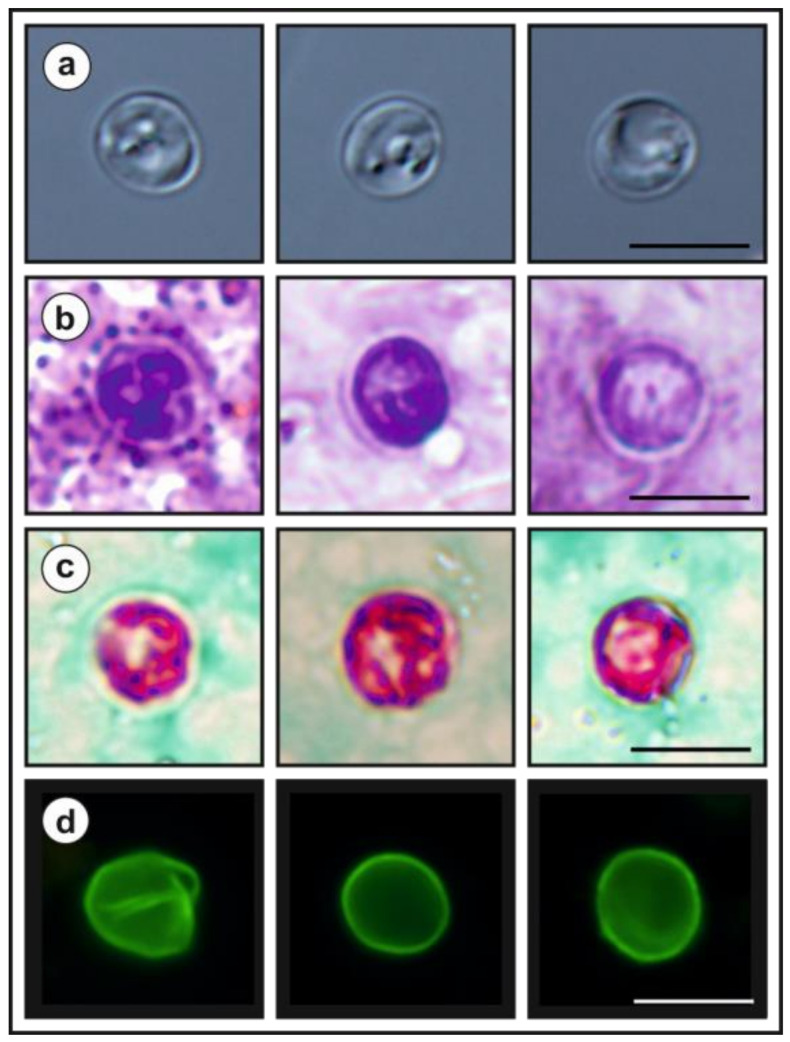
Oocysts of *Cryptosporidium myocastoris* n. sp. (**a**) in differential interference contrast microscopy, (**b**) stained by aniline–carbol–methyl violet staining, (**c**) stained by Ziehl–Nielsen staining, and (**d**) labelled with anti-*Cryptosporidium* FITC-conjugated antibody. Bar = 5 μm.

**Table 1 microorganisms-09-00813-t001:** The occurrence and genetic diversity of *Cryptosporidium* spp. in the present study detected by the amplification of small subunit ribosomal rRNA (*SSU*), actin, 70 kDa heat-shock protein (*HSP70*) and 60 kDa glycoprotein (*gp60*) gene fragments in the wild coypus (*Myocastor coypus*) in the Czech Republic and Slovakia. Oocysts were quantified by microscopy and reported per gram of faeces (OPG).

Country	Locality ^#^	No. of Positive/No. of Screened	ID of Positive Animal	Microscopical Positivity (OPG)	Genotyping at the Gene Loci			
					*SSU*	Actin	*HSP*70	*GP*60
Czech Republic	Planá nad Lužnicí (1)	1/11	41838	No	*C. parvum*	*C. parvum*	*C. parvum*	IIa
	Praha (2)	0/15	-	-	-	-	-	-
	Jihlava (3)	0/7	-	-	-	-	-	-
	Třebíč (4)	0/6	-	-	-	-	-	-
	Břeclav (5)	0/6	-	-	-	-	-	-
	Týnec (6)	0/15	-	-	-	-	-	-
	Lanžhot (7)	2/12	29639	No	*C. myocastoris*	*C. myocastoris*	*C. myocastoris*	XXIIb
			29370	No	*C. myocastoris*	*C. myocastoris*	*C. myocastoris*	XXIIa
Slovakia	Nové Zámky (8)	2/12	31467	No	*C. ubiquitum*	*C. ubiquitum*	*C. ubiquitum*	XIId
			31472	No	*C. myocastoris*	*C. myocastoris*	*C. myocastoris*	XXIIb
	Komárno (9)	0/3	-	-	-	-	-	-
	Šaľa (10)	1/5	31459	Yes (25,000)	*C. myocastoris*	*C. myocastoris*	*C. myocastoris*	XXIIa
	Dolný Ohaj (11)	0/7	-	-	-	-	-	-
	Topoľníky (12)	0/10	-	-	-	-	-	-
	Palárikovo (13)	1/6	-	-	-	-	-	-
	Nitrianský Hrádok (14)	0/5	-	-	-	-	-	-
	Dunajská Streda (15)	5/19	31123	No	*C. ubiquitum*	*C. ubiquitum*	*C. ubiquitum*	XIId
			31129	Yes (18,000)	*C. ubiquitum*	*C. ubiquitum*	*C. ubiquitum*	XIId
			31135	No	*C. ubiquitum*	*C. ubiquitum*	*C. ubiquitum*	XIId
			31136	No	*C. ubiquitum*	*C. ubiquitum*	*C. ubiquitum*	XIId
			31132	Yes (10,000)	*C. myocastoris*	*C. myocastoris*	*C. myocastoris*	XXIIa
	Vlčany (16)	0/6	-	-	-	-	-	-
	Diakovce (17)	0/1	-	-	-	-	-	-
	Lipové (18)	0/4	-	-	-	-	-	-

**#** Numbers of localities correspond with numbers in Figure 1.

**Table 2 microorganisms-09-00813-t002:** Size of *Cryptosporidium myocastoris* n. sp. oocysts recovered from naturally * and experimentally ^#^ infected nutrias (*Myocastor coypu*).

Isolate	Length (μm) Range (Mean ± SD)	Width (μm) Range (Mean ± SD)	Length/Width Ratio Range (Mean ± SD)
Nutria 31132 *	4.8–5.2 (5.02 ± 0.13)	4.7–5.0 (4.85 ± 0.10)	1.00–1.08 (1.04 ± 0.02)
Nutria 31459 *	4.8–5.3 (5.01 ± 0.14)	4.7–5.0 (4.81 ± 0.10)	1.00–1.06 (1.04 ± 0.01)
Nutria N0#	4.8–5.2 (5.00 ± 0.12)	4.7–5.0 (4.79 ± 0.09)	1.02–1.09 (1.04 ± 0.02)
Nutria N1#	4.8–5.3 (5.02 ± 0.14)	4.6–5.1 (4.85 ± 0.14)	1.02–1.07 (1.03 ± 0.01)

Note: Length and width of 30 oocysts from each isolate were measured under differential interference contrast at 1000× magnification, and out of these the length-to-width ratio of each oocyst was used to calculate.

## Data Availability

Not applicable.

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
