# Peer review of "Cryptosporidium myocastoris n. sp. (Apicomplexa: Cryptosporidiidae), the Species Adapted to the Nutria (Myocastor coypus)"

_microorganisms, 2021, doi:10.3390/microorganisms9040813_

Round 1

Reviewer 1 Report

This paper describes a new Cryptosporidium species from feral nutrias located in the Czech Republic and Slovakia.

The authors provide a comprehensive description on the characterisation of this new Cryptosporidium species which is supported by numerous methodologies (mostly well detailed, see comment below) and rigorous analyses of the results. Overall the manuscript is well written and suitable for publication.

Methodology to obtain OPGs needs to be explained. The cited paper Kvac et al (2007), does not fully explain or justify the method used to obtain OPGs. Method cited involves faecal smear, slide weighed, oocysts counted, OPG obtained. However, a faecal smear requires a tiny amount of faeces and therefore sensitivity is known to be low. How can a representative oocyst per gram of faeces be obtained on such a small sample size? Further to this the lack of sensitivity of a faecal smear should be considered when interpreting the microscopy results.

A few minor points are listed below.

Line 55 – “also call coypu’ should be “also called coypu”

Line 419 – Martino, Radman, Parrado etc (14) …..should be shortened to Martino et al (14) as shown elsewhere in the manuscript.

Line 470 – “Borowski et al reporting the elongation” – should be” Borowski et al reported the elongation”

Line 476 – “adaptation to the different part of….” Should be - “adaptation to a different part of…”

Line 480 – “None developmental stages” – Should be “No developmental stages”.

Lines 496-499 – should be deleted.

Author Response

Dear editor,

Enclosed you will find the revised version (R1) of our manuscript. The authors appreciate the thorough review and constructive comments to improve the manuscript. We have revised the manuscript text according to reviewer requests, and we have provided a response to each of the reviewers’ comments. During the final correction of English, the text was sometimes modified to improve readability.

Reviewer 1

This paper describes a new Cryptosporidium species from feral nutrias located in the Czech Republic and Slovakia.

The authors provide a comprehensive description on the characterisation of this new Cryptosporidium species which is supported by numerous methodologies (mostly well detailed, see comment below) and rigorous analyses of the results. Overall the manuscript is well written and suitable for publication.

Q1

Methodology to obtain OPGs needs to be explained. The cited paper Kvac et al (2007), does not fully explain or justify the method used to obtain OPGs. Method cited involves faecal smear, slide weighed, oocysts counted, OPG obtained. However, a faecal smear requires a tiny amount of faeces and therefore sensitivity is known to be low. How can a representative oocyst per gram of faeces be obtained on such a small sample size? Further to this the lack of sensitivity of a faecal smear should be considered when interpreting the microscopy results.

A1

Additional text was added to clarify that the OPG is estimated from the mass of the faecal smear. The exact text added is as follows:

The OPG was estimated from the total number of oocysts on the slide and the mass of the faecal smear (approximately 0.015 g).  

As the reviewer notes, the faecal smear method lacks sensitivity, which is a widely accepted limitation. Indeed, our results show that we detected fewer Cryptosporidium positive samples using this method than PCR-based methods, which is why we always included PCR for detection in addition to microscopy. Despite this limitation, we argue that there is considerable value in estimating OPG, particularly during experimental infections. The relatively tight error bars in infection intensity estimates, reported in Figure 6, suggests that they are reproducible, despite the small sample size.

A few minor points are listed below.

Q2 Line 55 – “also call coypu’ should be “also called coypu”

A2 Has been corrected.

Q3 Line 419 – Martino, Radman, Parrado etc (14) …..should be shortened to Martino et al (14) as shown elsewhere in the manuscript.

A3 has been corrected.

Q4 Line 470 – “Borowski et al reporting the elongation” – should be” Borowski et al reported the elongation”

A4 Has been corrected.

Q5 Line 476 – “adaptation to the different part of….” Should be - “adaptation to a different part of…”

A5 Has been corrected.

Q6 Line 480 – “None developmental stages” – Should be “No developmental stages”.

A6 Has been corrected.

Q7 Lines 496-499 – should be deleted.

A7 Has been deleted.

Reviewer 2 Report

Although the aim of the investigation submitted by Kezkova is not my study area, I think that this manuscript is very interesting due to reveal a new cryptosporidium species in the invasive species, Myocastra coypus.

The research is well structured and the results are very interesting.

However, before publishing this manuscript, English must be revised by an expert because in all text are present a lot of languages mistakes

Author Response

Dear editor,

Enclosed you will find the revised version (R1) of our manuscript. The authors appreciate the thorough review and constructive comments to improve the manuscript. We have revised the manuscript text according to reviewer requests, and we have provided a response to each of the reviewers’ comments. During the final correction of English, the text was sometimes modified to improve readability.

Reviewer 2

Although the aim of the investigation submitted by Kezkova is not my study area, I think that this manuscript is very interesting due to reveal a new cryptosporidium species in the invasive species, Myocastra coypus.

The research is well structured and the results are very interesting.

Q1 However, before publishing this manuscript, English must be revised by an expert because in all text are present a lot of languages mistakes

A1

The text of the corrected manuscript was revised by a native speaker, prof. John McEvoy.

Reviewer 3 Report

This manuscript describes the novel C. myocastoris, a species of the Cryptosporidium parasite that has adapted to infect Myocastor coypus. The authors do an admirable job demonstrating that the C. myocastoris parasite has a very narrow host range and does not infect a wide range of hosts as other Cryptosporidium species do. Their claim of a novel species is also well supported by the evidence. The manuscript could use some editing attention and there are some improvements to the figures and discussion suggested below.

Lines 49-54: The rationale behind grouping Cryptosporidium into two group is unclear. This might be a little clearer to the reader if the authors just explained that Cryptosporidium species differ their host range… some have a wide host range… others have a narrow host range.

Lines 68-69: This reference is from non peer-reviewed data and should not be included as supporting evidence.

Table 2: I’m not sure this needs to be in the main manuscript. Perhaps included as supplementary. It would also help to include the sizes of other Cryptosporidium species oocysts for comparison.

Table 3: Again, this feels like supplementary material to this reviewer.

Lines 485-491: I’m sure I understand the message here. The authors are just stating the distance between species. Any conclusions?

Lines 496-499: This might be some manuscript instructions accidentally left in.

General concerns/questions:

The authors demonstrate images from histological examination but do not include quantification of parasite burden.  The manuscript would benefit from a comparative quantification of parasites burden for the different sections of intestine.

Authors should include the primers that were used in this study in the methods so that the reader does not have to search through 4-5 references for them.

It is very interesting that the actin and HSP70 phylogenetic trees are so similar, while the SSU and gp60 is not. Cryptosporidium gp60 is highly polymorphic, so this is to be expected, but SSU should not be. This is likely because there are multiple SSU loci within the genome (C parvum has at least 4). This should be noted in the manuscript because it is difficult at best, and misleading at worst, to draw phylogenetic conclusions from a gene that exists in multiple copies. The Cryptosporidium field as a whole needs to address this, and I don’t expect the authors to do so in this manuscript, but it should definitely be prominently noted.

It would be nice to hear some speculation (based on phylogeny and the infection data here) of whether the authors believe Cryptosporidium myocastoris came over to Europe with the Coypu or adapted from another species. From the manuscript, I would guess that the authors would say the Cryptosporidium came over, but this wasn’t stated or included as a possible future study.  

The authors might want to carefully consider their keywords. ‘Cryptosporidium’ and ‘parasite’ are not included.

Author Response

Dear editor,

Enclosed you will find the revised version (R1) of our manuscript. The authors appreciate the thorough review and constructive comments to improve the manuscript. We have revised the manuscript text according to reviewer requests, and we have provided a response to each of the reviewers’ comments. During the final correction of English, the text was sometimes modified to improve readability.

Reviewer 3

This manuscript describes the novel C. myocastoris, a species of the Cryptosporidium parasite that has adapted to infect Myocastor coypus. The authors do an admirable job demonstrating that the C. myocastoris parasite has a very narrow host range and does not infect a wide range of hosts as other Cryptosporidium species do. Their claim of a novel species is also well supported by the evidence. The manuscript could use some editing attention and there are some improvements to the figures and discussion suggested below.

Q1

Lines 49-54: The rationale behind grouping Cryptosporidium into two group is unclear. This might be a little clearer to the reader if the authors just explained that Cryptosporidium species differ their host range… some have a wide host range… others have a narrow host range.

A1

This part has been corrected as follows: Concerning host specificity, some species of Cryptosporidium have a broad host range (e.g., C. parvum, C. meleagridis, C. baileyi and C. ubiquitum), whereas others are restricted to a narrow group of hosts (e.g., C. muris and C. andersoni) or a single host (e.g., C. wrairi) [13].

Q2

Lines 68-69: This reference is from non peer-reviewed data and should not be included as supporting evidence.

A2 The reference has been deleted.

Q3

Table 2: I’m not sure this needs to be in the main manuscript. Perhaps included as supplementary. It would also help to include the sizes of other Cryptosporidium species oocysts for comparison.

A3

We understand the reasons for including of Tables 2 into supplementary material. For reasons of readability, we would like to keep the structure of the manuscript as it is. The reader has the opportunity to look at the results immediately after reading them and does not have to look for them in supplementary. We also consider the inclusion of the tables directly into the text to be more comfortable for readers. If the the reviewer and editor insist on inclusion of tables as supplementary, we will comply with the request. We add new supplementary table A1 showing sizes of other Cryptosporidium species. 

Q4

Table 3: Again, this feels like supplementary material to this reviewer.

A4

The table was moved to the supplementary section

Q5

Lines 485-491: I’m sure I understand the message here. The authors are just stating the distance between species. Any conclusions?

A5

The following conclusion has been added. These results support the genetic uniqueness of C. myocastoris n. sp. and their status as a separate species of genus Cryptosporidium.

Q6

Lines 496-499: This might be some manuscript instructions accidentally left in.

A6

We are sorry, this text was erroneously left in. The text has been deleted.

General concerns/questions:

Q7

The authors demonstrate images from histological examination but do not include quantification of parasite burden.  The manuscript would benefit from a comparative quantification of parasites burden for the different sections of intestine.

A7

The text has been modified as follows:  Histology and SEM showed low infection intensity, with one or two developmental stages typically observed on an isolated villus in the posterior jejunum and ileum (Figures 7 and 8). This low infection intensity was consistent throughout the posterior jejunum and ileum.

Q8

Authors should include the primers that were used in this study in the methods so that the reader does not have to search through 4-5 references for them.

A8

The sets of primer have been added to the text as follows: For the SSU fragment, the primers 5´TTCTAGAGCTAATACATGCG3´ and 5´CCCATTTCCTTC GAAACAGGA3´ were used in the primary reaction, and the primers 5´GGAAGGGTTGTATTTATTAGATAAAG3´_and 5´AAGGAGTAAGGAACAACCTCCA3´ were used in the secondary reaction. For the actin fragment, the primers 5´ATCRGWGAAGAAGWARYWCAAGC3´_and 5´AGAARCAYTTTCTGTGKACAAT3´ were used in the primary reaction, and the primers 5´CAAGCWTTRGTTGTTGAYAA3_and 5´TTTCTGTGKACAATWSWTGG3´ were used in the secondary reaction. For the HSP70 fragment, the primers 5´GCTCGTGGTCCTAAAGATAA3´and 5´ACGGGTTGAACCACCTACTAAT3´_ were used in the primary reaction, and the primers 5´ACAGTTCCTGCCTATTTC3´and 5´GCTAATGTACCACGGAAATAATC3´ were used in the secondary reaction. For the gp60 fragment, the primers 5´ATAGTCTCCGCTGTATTC3´ and 5´GGAAGGAACGATGTATCT3´ were used in the primary reaction and the primers 5´TCCGCTGTATTCTCAGCC3´_and 5´GCAGAGGAACCAGCATC3´ were used in the secondary reaction.

Q9

It is very interesting that the actin and HSP70 phylogenetic trees are so similar, while the SSU and gp60 is not. Cryptosporidium gp60 is highly polymorphic, so this is to be expected, but SSU should not be. This is likely because there are multiple SSU loci within the genome (C parvum has at least 4). This should be noted in the manuscript because it is difficult at best, and misleading at worst, to draw phylogenetic conclusions from a gene that exists in multiple copies. The Cryptosporidium field as a whole needs to address this, and I don’t expect the authors to do so in this manuscript, but it should definitely be prominently noted.

A9

The text of the discussion has been modified as follows: For species-level differentiation of Cryptosporidium, the SSU marker has served well for more than 20 years [83]. However, Cryptosporidium, like the related apicomplexans Plasmodium and Eimeria, has divergent paralogous copies of the SSU gene [84-88]. Our previous work has shown that using only sequences of SSU to infer evolutionary relationships of Cryptosporidium may lead to erroneous conclusions [5,28,89]. Therefore, it is necessary to use other polymorphic loci, such as HSP70, actin or COWP genes, in phylogenetic analyses [5,85]. Although bootstrap support for the SSU tree was lower than for the actin and HSP70 trees in this study, C. myocastoris n. sp. formed a separate clade in SSU, actin and HSP70 ML trees, with the most closely related group comprising species such as C. parvum, C. cuniculi, and C. wrairi.

Q10

It would be nice to hear some speculation (based on phylogeny and the infection data here) of whether the authors believe Cryptosporidium myocastoris came over to Europe with the Coypu or adapted from another species. From the manuscript, I would guess that the authors would say the Cryptosporidium came over, but this wasn’t stated or included as a possible future study.  

A10 

The text has been modified as follows: In contrast, C. myocastoris n. sp. appears to have a narrow host specificity. The origin of C. myocastoris n. sp. in nutria in central Europe is difficult to elucidate without further studies, but it  may have been introduced into Europe with imported nutrias, similarly to the Cryptosporidium skunk genotype, which was likely introduced to Europe with eastern gray squirrels [59]. The specificity of C. myocastoris n. sp. for nutrias is supported by its presence in geographically isolated nutrias, its infectivity for nutrias under experimental conditions and the absence of any record of this species in any of the thousands of molecular epidemiological studies published in last two decades [45,50,60,61]. For these reasons, it is most likely that nutrias are the major host of C. myocastoris n. sp., although we cannot exclude the possibility that other host species have a role as major or minor hosts.

Q11

The authors might want to carefully consider their keywords. ‘Cryptosporidium’ and ‘parasite’ are not included.

A11

We used the keyword “parasite” instead of the keyword “new species”. The use of the word Cryptosporidium as keyword is not necessary because it is in the title of the article and the keywords should not match the words in the title.